# Improving Translation by Identifying Evidence for More Human-Relevant Preclinical Strategies

**DOI:** 10.3390/ani10071170

**Published:** 2020-07-10

**Authors:** Merel Ritskes-Hoitinga, Cathalijn Leenaars, Wouter Beumer, Tineke Coenen-de Roo, Frans Stafleu, Franck L. B. Meijboom

**Affiliations:** 1SYRCLE, Department for Health Evidence (section HTA), Radboud Institute for Health Sciences, Radboud University Medical Centre, 6500 HB Nijmegen, The Netherlands; 2Unit Animals in Science and Society, Department of Population Health Sciences, Faculty of Veterinary Medicine, Utrecht University, 3508 TD Utrecht, The Netherlands; Leenaars.Cathalijn@mh-hannover.de (C.L.); F.L.B.Meijboom@uu.nl (F.L.B.M.); 3Institute for Laboratory Animal Science, Hannover Medical School, 30625 Hannover, Germany; 4ProQR Therapeutics NV, 2333 CK Leiden, The Netherlands; WBeumer@proqr.com; 5Central Animal Facility, Leiden University Medical Centre, 2300 RC Leiden, The Netherlands; T.coenen@coenenconsultancy.nl; 6Ethics Institute, Faculty of Humanities, Utrecht University, 3508 TC Utrecht, The Netherlands; F.R.Stafleu@uu.nl

**Keywords:** animal-to-human translation, systematic review, translational strategies, ethics

## Abstract

**Simple Summary:**

To develop new medical treatments, animal studies are used. However, there are questions and concerns about the usefulness of preclinical animal research. These so-called translational success rates vary between 0 and 100% and no clear relationship has been established with possible predictive factors such as animal species or field of research. This paper presents the main results of a conference that was organised in November 2019 as part of a research project that focuses on ways to improve predictability of translation from preclinical research to clinical studies. Based on the conference results and the findings from the research project, we define four points of attention that are crucial in the search for improved translational success rates: (a) optimising the methods and design of studies; (b) incorporation of the complexity of the human patient in research; (c) start with the patient rather than existing animal models as the gold standard; and (d) more and better collaboration within the chain from funding to pharmacy. We conclude that this requires improved organization and use of procedures, as well as a change of attitude and culture in research.

**Abstract:**

Preclinical animal studies are performed to analyse the safety and efficacy of new treatments, with the aim to protect humans. However, there are questions and concerns about the quality and usefulness of preclinical animal research. Translational success rates vary between 0 and 100%, and no clear relationship has been found with possible predictive factors such as animal species or field of research. Therefore, it is not yet possible to indicate what factors predict successful translation. Translational strategies were therefore discussed at an international conference held in the Netherlands in November 2019, aiming to develop practical guidelines for more robust animal-to-human translation. The conference was organised during the course of a research project funded by the Dutch Research Council (313-99-310), addressing possible solutions for the low translational values that had been published for a multitude of animal studies in human health care. This article provides an overview of the project and the conference discussions. Based on the conference results and the findings from the research project, we define four points of attention that are crucial in the search for improved translational success rates: (a) optimising the methods and design of studies; (b) incorporation of the complexity of the human patient in research; (c) start with the patient rather than existing animal models as the gold standard; and (d) more and better collaboration within the chain from funding to pharmacy. We conclude that this requires improved organization and use of procedures, as well as a change of attitude and culture in research, including a consideration of the translational value of animal-free innovations and human-relevant science.

## 1. Background of the Conference

Research progress in the field of translational strategies was discussed at an international conference in the Netherlands in November 2019, aiming to develop practical guidelines for more robust animal-to-human translation. The conference was organised during the course of a research project funded by the Dutch Research Council (313-99-310), addressing possible solutions for the low translational values that had been published for a multitude of animal studies in human health care [1,2,3,4]. The project started in 2015 and aimed to examine the role of internal and external validity for translational success/failure. Lack of internal validity (internal validity refers to the result being accurate for the studied population) is a major issue hampering reliable scientific interpretation of animal studies, as many details of animal studies are not reported and basic measures like randomisation and blinding are not routinely included in the experimental design [5]. External validity refers to the possible generalization of the findings, e.g., across other situations, individuals, and species. In the current paper, external validity mainly refers to the possible generalization of results from animal studies to humans, also called translatability. Because of, e.g., the inherent biological differences between animals and humans, the relevance of the animal model, and the limited representativeness of animal samples, it is still unclear whether these translational hurdles can ever be “solved”. Therefore, the search for predictive animal-free models was also incorporated in the research project.

Two case studies, rheumatoid arthritis (RA) and cystic fibrosis (CF), were analysed to formulate best practices for building translational strategies. The project aimed to encompass steps and aspects of the entire research and development chain in health care, involving multiple stakeholders from academia, industry, and patient organisations.

At the start of the conference, Franck Meijboom presented the preliminary conclusions of the research project, including three areas for consideration that are key to achieving more successful translation from bench to bedside. The first is to improve scientific methodology. The basic principles of science, namely randomisation, blinding and appropriate experimental design, and statistical methods, need to be implemented much better, in order to achieve more reliable scientific results and to improve the interpretations of these results. The deficiency in fulfilling these basic methodological requirements for animal studies has been a matter of discussion for quite some time now, and the important question has been raised as to why there are no major improvements yet.

A second concern is the position of the human patient. One of the project’s hypotheses was that robust translational strategies ought to start from the patient’s perspective. To decrease the numbers of animals used, animal models are often quite specific and standardised; for example, an inbred mouse strain with all individuals having the same genetic background, with one specific gene that has been “knocked out” to study the effect of this single gene. This situation often does not match with the holistic problem(s) the patient is confronted with, such as having co-morbidities and interactions between multiple genes. Translation from animal studies to human patients, i.e., external validity, is then compromised, which we refer to as the “standardisation-translation paradox” [6]. It is claimed that the long-term aim of many animal studies is to contribute to human health and wellbeing. Therefore, health questions arising from the patient’s perspective should get a more central and leading role in order to focus on the most relevant research models that will provide answers to the most pressing health care issues.

Finally, the importance of an integrative approach to bring preclinical and clinical stakeholders together has been a central element in the research project. An investigation into institutional hurdles towards this integrated approach was carried out. This is essential, because the current situation resembles a fragmented landscape. The various parts in the chain perform their own studies with at most limited interactions with the other parts. The need for a systematic analysis of currently used animal models and the use of the resulting data within the research chain was clearly identified.

Next to these points as presented by Franck Meijboom, the one major question on the agenda of the conference was whether the idea of animal-to-human translation may perhaps have become outdated. There are serious concerns about the generic possibility of reliable translation from animal studies to humans [7,8]. On the other hand, successful translation may still be possible if (and only if) a paradigm shift takes place in the design and performance of preclinical research. Reflection on the currently used translational strategies is thus urgently needed, in order to determine more successful strategies for future research and development.

With this paper we aim to present an overview of the main results and discussion points from the conference and formulate four crucial factors that could lead to improving the translational value of preclinical (animal) research.

## 2. The Background of the Problem: The Conference Speakers’ Perspective

The problems in the translation from preclinical animal studies to clinical research presented above do not have one simple cause. The problem is multi-layered and different aspects are interrelated. Nonetheless, it is possible to distinguish four issues that affect the current problems: (a) flaws in methodology; (b) the lack of and uncertainty about the success rates and predictive value of animal models; (c) the need for a change in focus when choosing animal models; and (d) the changing moral position on animals. On each of these fields, experts were invited to present their research and thoughts.

John Ioannidis, Professor in Epidemiology at Stanford University, presented the challenges we are facing with respect to the major methodological shortcomings in current scientific practice. He stated that there are differences between various fields of research: physics shows the lowest degree of bias and psychology the highest. Biology also has a high degree of bias. The bias in biology and psychology is mainly related to the small power of the studies. In biomedical literature, 96% of the publications shows significant results [9], which indicates that negative and neutral results are mostly not published, leading to publication bias when integrating the evidence. Moreover, the translational pipeline is considered to “leak at all stages” [10]. Ioannidis expressed his opinion that animal testing can probably be reduced by 90% because of these shortcomings.

Reproducibility studies may help to overcome these problems. Reproducibility efforts and checks have become quite popular (e.g., [11]). Industry generally repeats published academic studies internally before relying on the results. They want to be sure before investing resources in research in the next stages of the pipeline. The Prinz et al. [11] study shows that two thirds of the studies could not repeat the initially published results, leading to halting further development of those substances. When taking reproducibility seriously, we should distinguish three types:Reproducibility of methods (repeatability): adequate descriptions/reporting of the methods to repeat the experimental and computational procedures as exactly as possible.Reproducibility of results (replication): produce corroborating results in a new study following the same experimental methods.Reproducibility of inferences: conclusions of similar strength; what is the evidence?

Ioannidis proposed several possible strategies for improvements at the level of research methodology. First, one should strive for large-scale collaborations and adoption of a replication culture. For example, the genetics field has been very successful in joining forces to share data and distribute work. Other good examples are the formation of consortia and multi-centre trials. Secondly, he stressed the importance of standardising protocols [2]. Thirdly, registration of research protocols and otherwise improving transparency are considered essential to improve methodology and reproducibility. Major progress in reporting of methods and results in clinical medicine has already been achieved due to mandatory registration of planned trials. Already initiated important steps comprise making raw data available on open-access platforms and publishing registered reports. Transparency also remains important in so-called “stealth research”. In spite of the claim that these new start-ups lead to high-speed innovation, one should be very careful with biomedical innovation outside the peer-reviewed literature. Fourthly, progress is still needed in training and education. There is particular room for improvement at the level of design of animal experiments and statistical training, but also with regard to achieving good institutional practices and for integrating these good practices into “real life”. Ioannidis pointed to the need for a change in the reward system in science, such as, e.g., the Lancet Reward campaign [12]. 

These general issues are equally important in translational research, but there the first step should be defining and clarifying what translational success means. There are many different definitions for translational success. Two crucial elements play a role in translation: (a) How robust is the individual study? In other words, what is the internal validity? (b) What is the external validity, in other words, to what extent can animal study findings be generalized and translated to humans? At this point, these questions remain to be answered.

Cathalijn Leenaars, postdoctoral researcher at Hannover Medical School, presented the results of a scoping review about the translational success and failure percentages derived from review articles [13]. The main research question of the scoping review was “What is the observed range of the animal to human translational success and failure rates within the currently available empirical evidence?” The original operational definition for translational success had been formulated as follows: “replication in a randomized trial in humans of statistically significant positive, negative or neutral results for the primary study outcome in animal experiments”. Translational failure was conversely defined as “not replicating the results of animal experiments in a randomized trial for the primary study outcome”. In practise, only a few references detailed the types of trials and experiments or the primary study outcomes. Therefore, the working definition had to be adapted, and translation was redefined as “the quantitative degree of correspondence between the results from trials in humans with results in animal experiments”. The results of the 121 included references in the scoping review demonstrated a variation in reported success percentages of translation from 0 to 100%. The number of publications increased over time during the period 1960–2019, but the percentages stayed within the same large range. It was not possible to predict the translation of animal studies to humans with any of the factors analysed: species, overall research field, study size, definition, and method of calculation of translational success. Larger reviews all focussed on adverse events and resulted in translational success rates below 50%. Further meta-research remains necessary to identify the key predictive factors in translational success.

Pandora Pound, Consultant at the Safer Medicines Trust UK, analysed the opinions scientists expressed in publications on translation in stroke [14]. The economic costs of one stroke patient is about 46,000 British pounds per year. The human and economic costs of stroke are a motivating factor behind research to find better treatments. Huge efforts are thus made in animal studies for finding new drugs to treat stroke, which have resulted in many successful new treatments in animal models. Despite these efforts, only one successful drug (tpa) has become available clinically. Worryingly, some of the drugs that were successful in stroke animal models resulted in a higher mortality in humans.

Pound examined the opinions that were expressed in 80 stroke papers on animal models and the poor clinical translation. Twenty-four papers suggested that a greater scrutiny of animal studies, including systematic reviews, would be needed. While the use of the STAIR (Stroke Therapy Academic Industry Roundtable) guidelines did lead to certain improvements in the quality of reporting, this did not lead to improved translation. The drug NXY-059 was developed according to the best preclinical and clinical guidelines; however, this did not result in clinical effectiveness.

The low translational success of animal studies warrants a deeper discussion on the lack of their usefulness and has highlighted the need for a culture and paradigm change. New movements have started to focus on animal-free innovations instead of animal studies. Examples are the white paper on Human Relevant Science UK 2020 (https://www.humanrelevantscience.org/white-papers/), the Transition Programme for Innovation without the use of animals (TPI) in the Netherlands (https://www.transitieproefdiervrijeinnovatie.nl/english), and the Environmental Protection Agency’s (EPA) goal to stop animal testing on mammals by 2035 (https://www.epa.gov/newsreleases/administrator-wheeler-signs-memo-reduce-animal-testing-awards-425-million-advance). Pound used the term “paradigm bridging” to refer to the use of in vitro and in vivo methodologies together, which is becoming increasingly popular. It is also feasible and perhaps even advisable to do proof-of-concept studies in humans before starting full clinical trials. As the current culture is still focused on the idea that animal studies are required, and also legally demanded when developing new treatments, it is quite a challenge to implement changes, even when there is sufficient evidence to show that this change is preferable. Frank (2005) [15] uses the terms psychological lock-in (for individuals) and institutional lock-in (for organisations) to illustrate this difficulty.

The presentation and discussion afterwards showed that the current paradigm is focussed on developing new therapies for diseases. This does not necessarily lead to improved health in society as a whole. On the contrary, it might even stimulate dependence on treatments by the health care system. By changing the focus of the research to prevention of diseases, a higher quality of life can be achieved at lower costs and less dependence on health care, especially in the case of lifestyle diseases. Increased focus on disease prevention is expected to substantially reduce the need for animal testing.

A final dimension is the ethical component that characterizes the debate on translational strategies. A low translational value is not merely a problem of methodology or design, it is also problematic due to the fact that this type of research entails the use of animals, whose moral position in society is changing.

Aiden Molavi, PhD student at Utrecht University, discussed the ethical position of animals in the current academic culture. In contrast to human volunteers participating in clinical research, animals cannot give informed consent for their participation. It is however important to not just see them as instrumental for achieving “higher goals”. In animal ethics, several arguments have been proposed to stress that animals are morally considerable for their own sake and that we—as humans—can have duties towards animals [16,17]. Nonetheless, these arguments and focus on values do not always match with the harm–benefit assessment that is mentioned in the EU-directive (2010). This suggests that a reassessment of the interests of relevant stakeholders is needed. Therefore, the question is whether animals in translational research can and should also be considered stakeholders. This is not only a pragmatic line of reasoning to bring the terms of ethics and the harm–benefit assessment in balance. If animals are to be considered stakeholders, then this has direct moral implications even up to the level that they deserve equal moral consideration to humans. From the perspective of stakeholder theory, human neonates and people with severe intellectual or learning difficulties are not able to give their own informed consent, and because of that, they are represented by custodians. Animals are in a comparable situation and the argument for representation by custodians can be made. This is important to provide animals and their interests an independent place in the search for translational strategies.

## 3. Further Views on How Translation Can Be Facilitated

Given the multi-layered character of the problem, the solutions to translational failure are equally complex. There is no one-size-fits-all solution. The project of which this conference was a part focused on four elements considered key for obtaining higher translational success: (a) integration of and cooperation within translational research, also including private partners; (b) improved education; (c) improved research methodology; and (d) giving patients a central position. Again, experts were invited to present their research and thoughts on these elements.

In the NWO project, Julia Menon, Master student at Utrecht University, performed a study to get a better grip on the key actors who are involved in translational research and identify the key facilitators and hurdles. Three levels at which facilitators and barriers occur are distinguished. First, the level of the experiment with a focus on methodology and design. Secondly, the institutional level that focus on (the lack of) incentives to address problems of low translational success. Thirdly, the chain or system level that focuses on interaction, communication, and collaboration between partners who are involved in the different stages of translational research. These three levels are interrelated and form an interconnected “translational knot”. The aim of the study was to identify the factors, actors, and structures that have an impact on translational success in drug development research. With the help of semi-structured interviews, several facilitators and hurdles were identified. Overall, three facilitators were identified. First, more awareness and use of guidelines for experiments to improve the quality of the study design and the experiment. Secondly, better interaction with all and debate amongst chain partners, including funding agencies, ethics committees, and journal editors. Due to the many partners involved, there can be conflicting requirements that can frustrate translational success rates. Finally, transparency and collaboration are key facilitators. All of this requires a critical reflection on the role of commercial incentives and competition. Furthermore, institutes need to develop and communicate a translational perspective, which includes or starts from the patients’ perspective.

The role of private partners in the search for translational strategies was highlighted by Wouter Beumer, director of Pharmacodynamics and Immunology at ProQR Therapeutics NV (the biotech company ProQR Therapeutics focusses on RNA therapies to treat rare inherited diseases). At the start of the NWO research project, it was planned to cooperate on improving translational strategies and to support drug discovery and the development of treatments for cystic fibrosis. The aim was to perform the “ideal” animal experiment as a case study to learn if optimal quality could lead to better translation. All the lessons that had been learned within the current NWO research project on translational strategies were incorporated into the actual planning, design, execution, and analysis of the “optimal” animal experiment. The word “ideal” had to be changed to “optimal”, as it was not viable to perform all parts of the experiment ideally completely in practice.

After this experiment had been completed, and despite successful early clinical trials [18,19], ProQR put the further development of a treatment for cystic fibrosis on hold. Because market developments for cystic fibrosis treatment were faster and overhauled these studies, the focus shifted to retinal diseases, including Leber’s congenital amaurosis type 10 caused by mutations in the CEP290 gene. A relevant transgenic mouse model for this retinal disease, with a similar mutation as in the human disease, turned out not to be a suitable model; it had no ocular phenotype and there was limited recognition of the human intronic mutation by the mouse spliceosome [20,21]. As an alternative, retinal organoids from human stem cells were developed. This led to the successful discovery and further development of a candidate drug (Sepofarsen) [22,23]. The human-based in vitro model turned out to be more informative on the pharmacodynamic properties of candidate medicines than the animal model.

The importance of the role of education was highlighted by Berent Prakken, vice-dean and director of the biomedical education centre at the Utrecht Medical Center Utrecht. He stressed the need for training by starting to show that translational scientists seem to be lost in “no man’s land”. There is a huge pressure to publish, which leads to exhaustion and feelings of depression. He brought the Eureka institute (eurekainstitute.org; Siracusa, Italy) to our attention; the institute was founded to bring patients, industry, clinicians, artists, and researchers together, who then work together for a week on translational issues. Eureka also organizes a summer school for young students (called Apollo) on doing the right research right. The results are quite positive, and participants acknowledge the necessity of these educational activities and enjoy taking part.

In the general discussion, the position of the patient in this current project and the “ideal” position for future translational projects was highlighted. The final element of the project’s search for solutions to the problem of translational failure is a more prominent focus on the human patient and his/her health. The NWO project aimed at identifying strategies that would lead to more successful translation from animal studies to humans, also through better involvement of patients. By performing systematic reviews for two case studies in the field of rheumatoid arthritis and cystic fibrosis, we expected to obtain information on decisive predictive factors. Patients were asked to participate in these systematic reviews, in order to assist in focusing on research questions that are most relevant to patients. In the course of the project, it became clear that an overall overview of translational percentages from animal studies to humans was lacking, and patients could thus not be informed reliably in what way animal studies had been contributing to improving treatment for their disease. This led to the need to perform a systematic scoping review to create the required overview [13]. Unfortunately, the scoping review could not determine which factors contribute to a higher predictive value.

Remarkably, the overview revealed that toxicological studies did not demonstrate higher translational predictivity as compared to efficacy and pharmacokinetic-pharmacodynamic (PKPD) studies [13]. Our hypothesis was that toxicological studies follow the Good Laboratory Practice (GLP) guidelines [24], which implies that they adhere to higher quality standards than efficacy and PKPD studies, which would result in better translation. While we cannot be sure of the methodological quality of the studies, these results might indicate that higher quality execution of animal studies is not associated with improved translation. This raises the important question of whether we should keep on doing and improving animal studies or redirect focus to animal-free innovations/more human-relevant science instead. Through formulating research questions together with patients, a better research focus and choice of model system is expected to come into reach.

In our project, patient participation was incorporated into the execution of systematic reviews; cystic fibrosis (*n* = 1) and rheumatoid arthritis (*n* = 3) patients reviewed the protocols before the start of the work, and the resulting publications before the first submission. For both case studies, the patients came up with additional research questions that we could answer by adding to our data extraction or performing additional analysis. They also contributed ideas for follow-up projects. Their pre-submission review of the publications improved the clarity of our work. The CF-patient even identified search terms that the authors, physician, and research specialist had not thought of, thereby improving our search strategy [25]. 

Outside our project, the James Lind Alliance helps patient organisations and funders to cooperate in so-called Priority Setting Partnerships (http://www.jla.nihr.ac.uk/about-the-james-lind-alliance/about-psps.htm). By asking patients for their priorities, they have defined the top 10 research priorities for various diseases. One of the remarkable examples is asthma, where patients indicated they did not wish new drug treatments, but preferred physiotherapy exercises in order to cope with breathing difficulties caused by asthma. As a consequence, a clinical trial was funded that was very successful and showed satisfying results for patients [26]. The beauty of this example is that the patient’s needs were fulfilled while at the same time avoiding the need for animal studies.

## 4. How to Proceed

During the conference, small group discussions were held on various topics associated with translational strategies. Participants gave input into what possible solutions could lead to better translation. In the following, the five categories of proposed solutions have been summarised.

First, the importance of *improving methodology* was also highlighted in the group discussion. The specific points that could be helpful at this level are as follows:ARRIVE guidelines for reporting should be enforced. The ARRIVE guidelines are currently endorsed by over 1000 journals. However, the checklist is mainly used as a simple box-ticking exercise, which is considered a “lip service” only, instead of actually improving the reporting of experimental details in publications.More attention to internal validity versus external validity. For correct interpretations of studies, internal validity needs to be guaranteed. Currently, publications of animal studies show large deficiencies in basic scientific methodology, compromising the internal validity. Because of species differences, external validity cannot be guaranteed; can animal models ever translate reliably to humans? Systematic reviews can be used to analyse internal and external validity. Furthermore, ethics committees and animal welfare bodies can actively contribute to improving internal validity of animal experiments, for example by demanding randomisation and blinding.By making peer review of research protocols a requirement, study quality will be increased.Education is essential to improve the appropriate use of scientific experimental methods

*Data sharing* was mentioned as another important element in the search for better translational success. This included the following points:There is a need for transparency on procedures and all data should be reported. Knowing how an experiment has been performed and being able to evaluate all gathered data is necessary to correctly interpret the results.Having all research data freely available in databases, can benefit quality control and allow re-use of those data to produce new results and insights. In, e.g., CERN (Conseil Européen pour la Recherche Nucléaire; the European Organization for Nuclear Research which manages a range of particle accelerator facilities), thousands of researchers are free to use all data in the database on fundamental physics and produce new results with “just” new calculations/methods.All negative and neutral results must be published and made available to the scientific community to prevent unnecessary repetitions of experiments. While many scientists assume that journals will not readily accept papers with negative and neutral results, several studies have shown that they do (e.g., [27]).

It has been furthermore concluded that translational problems have not merely technical causes but are also rooted in the research culture and the research system. A number of points can serve to *improve research culture* and the *research system*:The culture tends to complain about problems rather than identify solutions or alternatives. For instance, many researchers complain about lack of time, money, and resources, while these are not necessarily real concerns. In our project, we performed an “ideal” animal experiment (manuscript in progress), which was not more expensive and hardly more time-consuming than performing a “normal” experiment. This illustrates that another state of mind can make things happen.It is important to focus on methods and professional behaviour. Next to problems of questionable research practices, bias in research can be a hurdle for translational success. This is not easily addressed. As humans we are biased and since scientists are human, scientists can also be biased. Neither denial nor mere acceptance is a solution. Raising awareness of this issue is a start: know yourself and be aware of mechanisms, e.g., group pressure. At this stage, paying attention to this question in education is essential. Furthermore, indirect self-control can be a useful start: use methods such as preregistration and the use of standardized guidelines to reduce the risk of bias interfering with research quality.The current reward system is mainly linked to authorship of publications, and then especially in high impact factor journals, creating perverse pressures. Indeed, some have suggested that it can be beneficial to perform underpowered studies and publish incorrect conclusions [28]. In cases where randomization and blinding have not been performed, there is a higher chance for positive results, which may not reflect the real effect. This can result in a very undesirable situation, as unreliable results are published and can become influential, leading to unnecessary animal and human trials on “promising” new therapies. This can compromise trust in science. The reward system needs to change to prevent this type of issue, by rewarding other output, e.g., rewarding the quality of the work and decreasing the focus on the number and impact factor of publications.The culture and system of publishing output also has a negative impact on the search for improved translations. This starts with the link between publishing and getting new funding: researchers need to demonstrate a sufficient number of publications on their CV. For improving the system of research, the focus should be on quality instead of high pressure on the quantity of publications. Furthermore, publications on negative and neutral results should receive the same reward as those on positive results.Finally, innovations in the field of research funding are necessary. Funding is currently based on the successful treatment of a disease found in animal studies. Positive results in preclinical studies generate money for further research. However, many of these treatments do not work in humans, or may even cause harm. This asks for a more critical evaluation of this translational process and careful communication with other stakeholders and the general public. Furthermore, it is important to explore different funding schemes; for instance, to take measures to prevent the so-called Matthew effect—where the same (famous) research groups/individuals always get more funding, thereby reducing the chances for younger scientists. In Switzerland, new funding rules have already been installed to reduce the Matthew effect (http://www.snf.ch/en/funding/directaccess/innovations-research-funding/Pages/default.aspx#Information%20about%20already%20implemented%20innovations).

Furthermore, a change in *focus towards more clinically relevant research* can improve the translational success.

Animal models must undergo continuous scrutiny for their value and usefulness. Are animal models really reflecting the corresponding human disease? For instance, models of depression may not model the human disease at all, and should therefore be abandoned [29].It is important to acknowledge that an experimental animal is not the same as a patient. That may seem a platitude, but research models have to balance between the clinical heterogeneity due to the complexity of the individual patients and the need to test with standardised animals under standardised conditions. To deal with this so-called “standardisation-translation paradox” [6], it is essential to start designing research models incorporating the complexity of the patient. Human disease is often complex, e.g., because of co-morbidities. Increasing this complexity refers, e.g., to introducing standardised heterogeneity and multicentre preclinical trials. Integrated data analysis of animal and human studies by translational meta-analyses may then also help to overcome translational uncertainty.Specific funding schemes can shift the research focus. By providing more grants for animal-free innovations and human-centred models/human-relevant science, more research in those directions will be stimulated. As an example of human-relevant science, *n* = 1 microdosing in children has been performed to analyse what dosage is suitable for an individual child; thus, more personalised treatments can become a reality without the use of animals.Next to understanding diseases and developing better treatments, more focus on the prevention of diseases is desirable. Especially in the case of lifestyle-related diseases, more human research on prevention will contribute to the quality of life of humans and prevent diseases from occurring, thereby avoiding the need for animal studies to develop new therapies.

Finally, *participatory research* is a promising start.

The public should be more involved in research. More participatory models need to be designed and implemented. This will incorporate public opinion into science, while at the same time improve the sharing of research data with the public. The public at large can have an influence on science and learn where the money from taxpayers and charity funds is spent in research, which will corroborate trust and reputation.A central role for patients is envisaged, as much research has the long-term perspective of helping patients. It is only logical that patients are being asked to give input about what is important to them.

## 5. Conclusions

We have given an overview of the discussions that were held on the final conference of our research project on improving animal-to-human translational strategies (NWO-MVI 313-99-310). Our NWO-MVI project has identified several factors that may contribute to translational success and has provided ideas for potential solutions.

Based on the results from this project and the discussions held during the conference, the search for improved translational success rates should begin by (a) optimising the methods and design of studies. This entails a careful evaluation and selection of (animal) models. Further meta-research remains necessary to determine if a sufficient degree of predictive certainty can be reached by improving translational strategies through the use of animal studies, or whether the focus should shift to animal-free alternatives. A case study on Leber’s congenital amaurosis type 10 demonstrated that by using a human-based in vitro model, better information on pharmacodynamic properties of candidate medicines were obtained than through the transgenic animal model. Then, (b) it is essential to start from the patient with all the encompassed complexity. This should help to deal with the so-called “standardisation-translation paradox” [6] and to formulate relevant research questions. Giving patients a central position may seem challenging. However, our experiences are positive. Based on ethical duties towards human health and protection of animal interests, we need to take this step. Finally, (c) translational strategies require more and better collaboration within the research chain. This is necessary, because the translational pipeline currently shows “leakage” at various stages, e.g., animal studies aiming to inform first administration to humans are not finished or fully analysed before the start of human studies [14,30,31], and because the level of complexity requires collaborations between several disciplines. A high-quality methodology is required throughout the pipeline to allow for meaningful and reliable interpretations of the results. 

Implementation of these factors requires improved organization, education, and procedures, but also a change in attitude and culture. We sincerely hope this conference has contributed to further discussion on how to achieve improved translation.

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
