# Peer review of "Improving Translation by Identifying Evidence for More Human-Relevant Preclinical Strategies"

_animals, 2020, doi:10.3390/ani10071170_

Round 1
Reviewer 1 Report
I have a declaration of interests:
- I know several of the authors of this manuscript
- I was present at the conference reported in the manuscript
- The organisers, some of which authors of this manuscript, paid for my travel expenses to be present at said conference.
I do not think this disqualifies me as a reviewer, but I leave it to the editor to mind the possibility of my inherent bias when taking this review into consideration. In full transparency, I decided to this review non-anonymously.
This manuscript reports the conclusions of a conference that brought together specialists in laboratory animal science, medical research, and comparative medicine, that provides a good contribution to the discussion of the translational value of preclinical research.
Some points, however, need some clarification.
The first point is that some authors have made a distinction between ‘preclinical research’ – i.e. to inform clinical studies on the safety and efficacy of candidate drugs and other therapies and ‘proof-of-concept, or fundamental research (see, for example, Ludolph et al 2010 DOI: 10.3109/17482960903545334). It is not clear whether the authors indeed have this distinction in mind and are focusing specifically in “preclinical research”, stricto sensu, or are referring to all animal research, as a whole, and therefore some clarification on this matter is in order.
One of the reasons why this matters is because methodological standards are higher in animal studies designed to inform clinical studies, i.e. stricto sensu preclinical studies (see for example Fernandes et al 2019 doi: 10.1136/bmjos-2018-000016) and because the informative value of animal models may be higher in basic biology studies, though this is hard to quantify.
The authors mention that translational success varies between 0% and 100%, which is hardly informative. It would be more informative to report how translational success rates indeed vary with field and species, as they recently reported in Leenars et al 2019 (DOI: 10.1186/s12967-019-1976-2) and had also been previously described (see Kola and Landis 2004 DOI: 10.1038/nrd1470)
I would also want to point out that authors sometimes use the term “external validity” as synonymous with translatability. While some literature does use the term in as similar manner, external validity is more akin to reproducibility, as it relates to the range of settings for which an experiment’s results holds true, which might be other laboratories, other non-human species, other strains, and not necessarily human patients. Indeed, one often sees external validity as the generalisability of results from a sample of humans to a larger population, so its application as being specific to animal to human translation is inaccurate. So it would be best, to avoid ambiguity, to use “translatability”, when authors are referring to how outcomes in animal models can predict outcomes in humans.
Further comments:
In line 56 the level of detail in reporting is associated with internal validity, when it has more to do with the ability to correctly interpret and replicate a study.
In lines 78-84, a point is made on the existence of a standardisation-translation paradox. Although I fully agree that animal models should recapitulate as much as possible the aetiology of human disease and experiment conditions should resemble the clinical setting (e.g. providing palliative care to animals) to improve their construct validity, the issue with standardisation is that it affects external validity, in the sense I referred to above (e.g. see Voelkl et al 2018 DOI: 10.1371/journal.pbio.2003693, or Richter et al 2009 DOI: 10.1038/nmeth.1312).
In line 109, please change “changing moral position of animals” to “changing moral position ON animals”.
In lines 118-120 we find a very bold statement from Ioannidis that “animal testing can probably be reduced by 90%”, but these numbers have no grounding on any evidence from data (and the paper cited immediately before does not even have the word “animal“ in it), and the authors are advised to leave it out. Ioannidis has been wrong before (and more so now, in the wake of his failed predictions on the current pandemic).
In lines 127-131, “reproducibility” is used to mean different things, but “repeatability”, “replicability” and “reproducibility” have precise meanings, even though they are often – and wrongly – used interchangeably. Why not use a new 3Rs formulation?: “repeatability of methods, allowed by their detailed description”, “replicability of results in new studies following same experimental methods”, and “reproducibility of inferences, by other teams, and other experimental approaches”?
In lines 144, it is argued that improvement is needed at the level of statistical training, but the most pressing matter is educating researchers in experimental design, especially the design of animal experiments (this is developed by Derek Fry 2014 10.1093/ilar/ilu031).
In lines 146-148, I do not understand how increased funding in “Blue Sky research” (which I fully support) will improve in any way the reproducibility and translatability of animal research.
Line 153, again, “external validity” as synonymous to “translatability”
Line 168 – Of course translatability ranges between 0% and 100%, so more detail is warranted on how it varies between fields, as well as with species, experimental conditions, and methodological quality.
Line 198-203 – I expect authors to be aware of the ethical issues raised by their proposal to focus on prevention (always necessary, of course, but as a complement), as a means to avoid animal research on the so-called lifestyle diseases (see Lund et al 2014 10.1136/medethics-2011-100368), and I advise them to acknowledge them.
Line 274 – Please check if “doing the right research right” is a typo, or it is written as intended.
Line 289 – Please provide a reference on what is GLP
Line 293 – Please add “of” between “question” and “whether”
Line 331 – “Ethics” is capitalized.
Lines 334-336 – This could be merged into the first bullet point.
Line 351 – Please consider changing “next it has been concluded” to “It has been furthermore concluded”
Line 381-382 – I believe it is more appropriate to refer to negative and positive results, as opposed “data”. Also, “neutral results” can be used interchangeably with “negative results”, rather than meaning something else.
Line 385 – Please change “generates” to “generate”.
Line 385-386 – Please consider changing “Many of these treatments do however not work in humans” to “however, many of these treatments do not work in humans”
Line 414-416 – It is not clear how this approach would prevent animal use at the preclinical stage
Line 434-436 – One approach is retrospective comparison of results from clinical trials of drugs available in the market and outcomes of preclinical studies on the same drugs (eg see Varga et al 2015 DOI: 10.1111/obr.12278)
Lines 438-441 – As I mentioned previously, this could also include adding palliative care to animals (e.g. to improve feeding and hydration)
Reviewer 2 Report
This paper is the report of an important conference on an important topic - namely, improving the methodological quality of pre-clinical, animal research intended to inform the development of treatments for humans. The working meeting follows a pioneering conference on this topic held in Nijmegen some years ago. Although I believe that a report of the November 2019 conference should be published, I think that the current draft would be more likely to be read widely if it was shortened and redrafted. Here are some suggestions that the authors and editors may wish to consider.
I suggest preparing a new draft focussing on the element providing the main 'take home messages' from the conference, namely, Section 4. An alternative would thus consist of:
Para 1 of 'Section 1. Background of the Conference'
All of 'Section 4. How to proceed', with an expanded account of the process that resulted in the bullet points (this is the Methods Section of the conference).
All of 'Section 5. Conclusion'
All of the rest of the paper - Sections 1 (except first para), 2, and 3 - could be presented as an Appendix for those readers who wish to consider the details provided in these sections.
The quality of the English in the draft is very variable, with that in the first half needing more attention than that in the second half, most of which is very good.
I hope these suggestions will be helpful.
Author Response
This paper is the report of an important conference on an important topic - namely, improving the methodological quality of pre-clinical, animal research intended to inform the development of treatments for humans. The working meeting follows a pioneering conference on this topic held in Nijmegen some years ago. Although I believe that a report of the November 2019 conference should be published, I think that the current draft would be more likely to be read widely if it was shortened and redrafted. Here are some suggestions that the authors and editors may wish to consider.
AU: It goes without saying that we appreciate the positive comments and that the conference report should be published. After consulting with the editor, the original sequence of the text was maintained, and the text was tightened. The English has been revised.
I suggest preparing a new draft focussing on the element providing the main 'take home messages' from the conference, namely, Section 4. An alternative would thus consist of:
Para 1 of 'Section 1. Background of the Conference'
All of 'Section 4. How to proceed', with an expanded account of the process that resulted in the bullet points (this is the Methods Section of the conference).
All of 'Section 5. Conclusion'
All of the rest of the paper - Sections 1 (except first para), 2, and 3 - could be presented as an Appendix for those readers who wish to consider the details provided in these sections.
The quality of the English in the draft is very variable, with that in the first half needing more attention than that in the second half, most of which is very good.
I hope these suggestions will be helpful.
Reviewer 3 Report
This report highlights a relevant problem in preclinical animal research: the poor reproducibility and translatability of applied animal research. All and all, the project has led into a conference report, and the authors were able to identify some factors that could contribute to translational success from animal studies to human patients. Poor translatability is generally known to be a multifactorial problem and various aspects are interrelated. Based on the results of the project and the discussions during the conference, the authors of this report proposed ideas for potential solutions for improved translational success rates including 1) optimization of methodology and design of animal studies in order to be able to make meaningful and reliable interpretations of scientific results, 2) embedding the complexity of human patients into preclinical research, 3) a reversed approach to translation (to answer the research question from patient's perspective to animal models), and 4) a search for translational strategies with better cooperation within the chain from funding to pharmacy in order to avoid leakage in the translational pipeline. The authors also made a call for a change in attitudes and culture in research.
To better guide readers of this publication who may not be fully familiarized with animal research, including medical and veterinary clinicians, the authors could make a clear distinction of what kind of preclinical studies they are referring to in this report. It is obvious that the report refers to translational animal research, but animals used in preclinical research may include translational (applied) studies as well as safety and efficacy studies, and maybe also animals used in veterinary research. Not all preclinical studies are aimed at finding a translational answer, and a clear definition of what they refer to preclinical research could improve the overall value of the manuscript. As a consequence, the title may also be reviewed to better express the main contents of the manuscript.
The abstract is clear and concise, the introduction is balanced, and appropriate references have been included. The authors made a call for a change in attitudes and culture in research, but this seems to be very general and open, and perhaps specific solutions for this may need to be proposed in future studies. The discussion of this manuscript was otherwise focused and appropriate.
Author Response
We appreciate the positive comments made.
The description of preclinical and translational research has been made more precise.
Reviewer 4 Report
This report presents the results of a conference held in the Netherlands designed to assess practical guidelines for animal-to-human research. the authors provide a well-written, detailed, and thought provoking assessment of translational research that resulted from this conference. I find this report to be particularly interesting in terms of challenging current paradigms, and I believe the results of this conference will be of great interest to this journal's readership.
I have no suggestions for improvement of the manuscript because it is so well done.
Author Response
It goes without saying that we really appreciate the positive comments given.